

# Key derivation function: key-hash based computational extractor and stream based pseudorandom expander

Chai Wen Chuah[1], Nur Ziadah Harun[2] and Isredza Rahmi A. Hamid[2]

[1] Guangdong University of Science & Technology, Dongguan, China
[2] Universiti Tun Hussein Onn Malaysia, Batu Pahat, Malaysia

## ABSTRACT

The key derivation function is a specific cryptographic algorithm that transforms private string and public strings into one or more cryptographic keys. The cryptographic keys are essential for protecting electronic data during transmission on the internet. This function is designed based on a computational extractor and pseudorandom expander and is typically constructed using various cryptography ciphers such as stream ciphers, keyed-hash message authentication codes, and block ciphers. Having secure and efficient key derivation function designs is essential in the development of numerous security systems. A vulnerable key derivation function could potentially give attackers the ability to compromise an otherwise secure cryptosystem. This research proposes a different approach by combining two different cryptography ciphers to develop key derivation functions. The findings demonstrate that a computational extractor utilizing keyed-hash message authentication codes and a pseudorandom expander using stream ciphers maintain the highest level of security while also providing efficiency benefits in terms of execution time compared to existing key derivation function schemes.

## INTRODUCTION

Key derivation functions are commonly applied in protocols like Transport Layer Security (*Rescorla, 2018*) and Host Identity Protocol (*Moskowitz et al., 2015*), making it crucial to develop secure key derivation function proposals to protect electronic data transmitted over insecure channels (*Italis, Pierre & Quintero, 2023*; *Muthakshi & Mahesh, 2024*; *Venčkauskas et al., 2024*; *Saif, Migliorini & Spoto, 2024*). In recent years, key derivation function proposals have typically been structured in two phases, involving a computational extractor and a pseudorandom expander (*Krawczyk, 2010*; *Barker, Chen & Davis, 2020*; *Chuah, Dawson & Simpson, 2013*; *Saran, 2024*). This two phases approach allows researchers to separately design and analyze the security of the extractor and expander components. The key derivation function is a cryptographic algorithm takes arbitrary length of private string and public strings and generates one or more pseudorandom

Corresponding author
Chai Wen Chuah,
aiwenchuah@gmail.com

cryptographic keys. The entropy of the private string determines the security of the key derivation functions.

Existing key derivation function schemes has keyed-hash message authentication code based key derivation function (*Krawczyk, 2010*), block cipher key derivation function (*Barker, Chen & Davis, 2020*), and stream cipher key derivation function (*Chuah, Dawson & Simpson, 2013*; *Chuah & Koh, 2017*). Keyed-hash message authentication code and block ciphers produce a fixed-length output from a variable-length input, requiring modification when the desired cryptographic key length is not an exact multiple of the output block size. This can result in wasted bits. To address this issue, a key derivation function based on stream ciphers has been suggested, allowing for the generation of cryptographic keys of any length without discarding excess bits. It should be noted that key derivation functions based on keyed-hash message authentication code offer higher security compared to those based on stream ciphers. The security of key derivation functions is dependent on the underlying ciphers, with keyed-hash message authentication code using SHA512 offering a higher brute force complexity of $2^{512}$ and collision complexity of $2^{256}$, while Trivium has a brute force complexity of $2^{80}$ and collision complexity of $2^{40}$.

To date, the key derivation functions scheme which consists of the computational extractor and the pseudorandom expander are constructed using the same cryptographic primitives. However, this approach has limitations, such as the block ciphers and the keyed-hash message authentication code can only produce fixed blocks length of cryptographic key and are slower, but with better security. On the other hand, stream ciphers can generate arbitrary length of cryptographic key, execute faster but offer lower security.

## Our contribution

In this article, we introduce the development of a keyed-hash based computation extractor and a stream-based pseudorandom expander. We denoted this design as HMSKDF. The cryptographic primitive that is constructing the keyed-hash based computation extractor is the keyed-hash message authentication code that utilizes the secure hash algorithm with an output block size of 512 bits. For the stream-based pseudorandom expander, the cryptographic primitive is Trivium. The output for HMSKDF is arbitrary length. The research will focus on evaluating the security, software, and hardware performance of this alternative approach. Overall, our proposed HMSKDF provides a significant advancement in the field of key derivation functions, offering better security for ensuring the pseudo randomness of the generated cryptographic keys and eliminating bits wastage.

## Organization of the article

We provide the theoretical definition of key derivation function in 'Key Derivation Function'. 'The proposed HMSKDF Scheme' presents the HMSKDF scheme, the security analysis for the proposed scheme, the software performance evaluation in term of execution time and hardware performance. 'Conclusion' concludes the article.

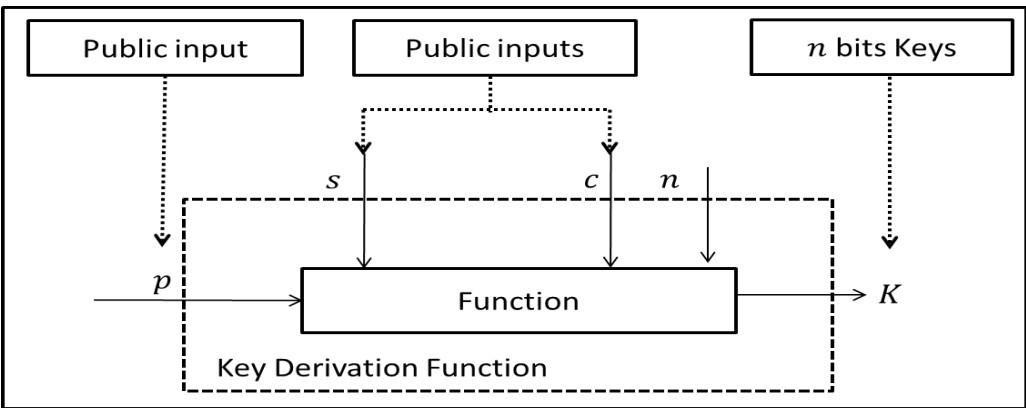

**Figure 1** **General key derivation function.**

## KEY DERIVATION FUNCTION

Key derivation function is a cryptographic function. It generates arbitrary length $n$ of cryptographic key $K$ from public string $p$, public salt $s$ and public context information $c$ as shown in Fig. 1.

Definition 1. Key derivation function is defined as *Krawczyk (2010)*, *Chuah, Dawson & Simpson (2013)* and *Saran (2024)*:

$$K \leftarrow (p, s, c, n) \tag{1}$$

- $p$ is private string, it is chosen from the public string space $\beta$, such that $p \in \beta$ with the length $l^p$ and probability distribution $\mathfrak{P}$.
- $s$ is a public random string consisting of salt space $\tau$, such that $s \in \tau$ with the length $l^s$ and probability distribution $\mathfrak{S}$.
- $c$ is a public context string consisting of context space $\omega$, such that $c \in \omega$ with the length $l^c$ and probability distribution $\mathfrak{C}$.
- $K$ isa pseudorandom cryptographic key.
- $n$ is a positive integer.

Definition 2. (Computational extractor) (*Krawczyk, 2010*). Set spaces of private string $p$ with $m$ min-entropy is $\beta$ and public random string $s$ is $\tau$. A computational extractor is defined as:

$$Extracor : \{0,1\}^{l^p} x \{0,1\}^{l^s} \rightarrow \{0,1\}^{l^{PKR}}. \tag{2}$$

The computational extractor is known as $(m, a_X, q_X, \varepsilon_X)$–min-entropy computational extractor if for all probabilistic polynomial time adversary $a_X$ who makes $q_X$, queries to the *Extractor*. The probability for the probabilistic polynomial time adversary may distinguish a string of the same length, either the string is the derived *PKR* from the *Extractor* or the string is just a random string is not larger than $\left(\frac{1}{2} + \varepsilon_X\right)$, where $\varepsilon_X$ is deemed negligible.

Definition 3. (Pseudorandom expander) (*Krawczyk, 2010*). Known public context string set space $\omega$. A pseudorandom expander is defined as:

$$Expander : \{0,1\}^{l^{PKR}} x \{0,1\}^{l^c} \rightarrow \{0,1\}^n. \tag{3}$$

The pseudorandom expander is known as $(a_Y, q_Y, \varepsilon_Y)$-pseudorandom expander if for all probabilistic polynomial time adversary $a_Y$ who makes $q_Y$ queries to the *Expander*. The probability for the probabilistic polynomial time adversary may distinguish a string of the same length, either the string is the derived cryptographic key from the *Expander* or the string is just a random string is not larger than $\left(\frac{1}{2} + \varepsilon_Y\right)$, where $\varepsilon_Y$ is deemed negligible.

## CAM security model

The security objective of a key derivation function is to ensure that the values of private string $s$ is unknown and public string c is known. The cryptographic key K generated by the key derivation function is indistinguishable from truly random binary strings of the same length. *Koh & Chuah (2020)* proposed the robust security model for key derivation function, namely CAM. The CAM serves as the formal security framework for evaluating the security of key derivation function. The CAM security model employs modern cryptography security proof technique, involving an indistinguishability game between two players known as the challenger and adversary. The indistinguishability game is performed in polynomial time $t$. The adversaries with the capability of influencing all the inputs of the key derivation function to conduct the bit-flipping attack, the objective is to identify any weaknesses in the key derivation function within polynomial time.

Definition 4. (CAM-secure) (*Koh & Chuah, 2020*). Key derivation function is $(m, t, q, \varepsilon)$ CAM-secure if for all probabilistic polynomial time adversary is making $q$ queries $(q < |\beta| \times |\gamma|)$ to the key derivation function with chosen bit position of the private string $s$. The adversary can win this indistinguishability game with probability not greater than $\left(\frac{1}{2} + \varepsilon\right)$, where $\varepsilon$ is deemed negligible.

1. Challenger chooses $p \leftarrow \beta$.
2. For $i = 1, \ldots, q' \leq q$,
   (a) Adversary chooses the bit position $z_i$ of p.
   (b) Adversary chooses $s_i \leftarrow \tau$ and $c_i \leftarrow \omega$.
   (c) Adversary sends $z_i$, $s_i$ and $c_i$ to challenger.
   (d) Challenger generates $K_i \xleftarrow{t_i} KDF(p, s_i, c_i, n)$, $t_i$ is time that used to generate the $K_i$.
   (e) Challenger sends $K_i$ and $t_i$ to adversary.
3. Adversary chooses $z$, $s \leftarrow \tau$ and $c \leftarrow \omega$, subjected that $(\{z, s, x\} \notin \{z_i, s_i, c_i\}, \ldots, \{z'_q s'_q, c'_q\}$. Adversary sends $z$, $s$ and $c$ to challenger.
4. Challenger chooses $b$ randomly, $b \xleftarrow{R} \{0,1\}$.
   (a) If $b = 0$, challenger generates $K' \xleftarrow{t_b} KDF(p, s, c, n)$, $t_0$ is the execution time that used to generate the cryptographic key $K'$.
   (b) Else challenger generates random $K' \xleftarrow{R, t_b} \{0,1\}^n$, $t_1$ is the random time that used to generate the random $K'$.
   (c) Challenger sends $K'$ to adversary.

5.  Continue $q - q'$ queries, follows the step 2 subjected $\{z_i, s_i, c_i\} \notin \{z, s, c\}$.
6.  Adversary win the game if $b' = b$.
    (a) If adversary output $b' = 0$, then adversary believes $K'$ is derived cryptographic key $K$ from key derivation function.
    (b) Else adversary output $b' = 1$.

## Hash-based message authentication code based key derivation function

*Krawczyk (2010)* proposed hash-based message authentication code based key derivation function (HKDF). The HKDF is constructed using computational extractor and pseudorandom expander. The HKDF has been proven to be CAM-secure (*Koh & Chuah, 2020*). The computational extractor and pseudorandom expander of HKDF are defined as:

$$PRK \leftarrow \mathfrak{F}((s \bigoplus opad) \| \mathfrak{F}((s \bigoplus ipad) \| p)) \tag{4}$$

$$K(i+1) \leftarrow \mathfrak{F}\left(\left(PRK \bigoplus opad\right) \| \mathfrak{F}\left(\left(PRK \bigoplus ipad\right) \| K(i) \| c \| i\right)\right) \tag{5}$$

- $\mathfrak{F}$ is hash function, either secure hash algorithm 1(SHA1) or secure hash algorithm 2 (SHA2).
- $\bigoplus$ is exclusive OR.
- $\|$ is string concatenation.
- opad is outer padding, is formed by repeating the byte 0x36.
- ipad is inner padding, is formed by repeating byte 0x5c.
- $1 \le i < \mathfrak{L}, \mathfrak{L} = \lceil \frac{n}{l^{PRK}} \rceil$.

The length $s$, $l^s$ must equal with $l^{PRK}$. Therefore, $s$ is hashed using $\mathfrak{F}$, if $l^s > l^{PRK}$. Or, $s$ is padded with zero until $l^s$ is equal with $l^{PRK}$, if $l^s < l^{PRK}$. The $l^{PRK}$ issame with the length of hash digest (SHA1 or SHA2). The first $n$ bits derived cryptographic key $K = K(1) \| \ldots \| K(\mathfrak{L} - 1)$ is utilized as cryptographic keys, while the remaining bits is discarded.

One benefit of HKDF is its ability to handle inputs of arbitrary length, but a drawback is that it generates fixed-length blocks and discards any excess bits, causing wastage.

Definition 5. (HMAC-PRF) (*Hirose, 2019*; *Gaži, Pietrzak & Rybár, 2014*; *Naik & Singh, 2024*). A hash-based message authentication code, $\mathfrak{F} : \mathbb{K}x\mathbb{D} \rightarrow \mathbb{R}$, with key of $\daleth \in \mathbb{K}$. A keyed-function with specific to the input length, such that $G : 0, 1^c x 0, 1^{b*} \rightarrow 0, 1^c$ is $(\varepsilon, t, g, l)$-HMAC-PRF secure, if all probabilistic polynomial time adversary $t$, making at most $q$ queries, each of length at most $l$ (the $b$bits block), a $R : 0, 1^{b*} \rightarrow 0, 1^c$ and a uniformly random key $\mathbb{K} \leftarrow 0, 1^c$, therefore, $\Delta^{\text{Adversary}}(G_{\mathbb{K}}, \mathbb{R}) \le \varepsilon$, where $\varepsilon$ is deemed negligible.

## Stream cipher based key derivation function

*Chuah, Dawson & Simpson (2013)* proposed a stream cipher based key derivation function (SCKDF). The SCKDF has been proven not CAM-secure (*Koh & Chuah, 2020*). The computation extractor of SCKDF is defined as:

$$PRK \leftarrow \mathfrak{S}\left[\left[\left(p^1 \bigoplus s\right) \bigoplus (p^2 \| p^3)\right] \bigoplus \cdots \bigoplus \left(p^{l-1} \| p^l\right)\right]. \tag{6}$$

The inputs for pseudorandom keystream generator ($\mathfrak{S}$) consist of key and initial vector. The length for the key, we denoted as $\mathfrak{l}^{sk}$. The length for the initial vector, we denoted it as $\mathfrak{l}^{iv}$. These inputs are substituted with the input pairs of key derivation function $p$ and $s$ (*Italis, Pierre & Quintero, 2023*). If $s$ is null, $p$ is divided into $\mathfrak{l}^{sk} + \mathfrak{l}^{iv}$ per block. If $s$ is not null, the length of $s$ is suggested to be same with $\mathfrak{l}^{sk}$ and $p$ is divided into $\mathfrak{l}^{iv}$ per block. If $\mathfrak{l}^{p}$ is greater than $\mathfrak{l}^{iv}$, the first block's length of $p$ is $\mathfrak{l}^{iv}$. The remaining block's length of $p$ is $\mathfrak{l}^{sk} + \mathfrak{l}^{iv}$. The pseudorandom keystream generator executes the entire blocks $p^{\mathfrak{l}}$ of $p$. Then, outputs $PRK$. The length of $PRK$, $\mathfrak{l}^{PKR}$ is equal to the key length of pseudorandom keystream generator for pseudorandom expander phase $\mathfrak{l}^{sk}$.

The pseudorandom expander of SCKDF is defined as:

$$K \leftarrow \mathfrak{S}\left[\left[\left(PRK \bigoplus c^{1}\right)\bigoplus c^{2}\right]\bigoplus \cdots \bigoplus c^{\mathfrak{l}}\right]. \tag{7}$$

The input for pseudorandom keystream generator (*Gaži, Pietrzak & Rybár, 2014*) is substituted with the input pairs of key derivation function $PRK$ and $c$ (*Chuah, Dawson & Simpson, 2013*). The public string $c$ can be null or not null. If $c$ is null, $c$ is $0^{\mathfrak{l}^{iv}}$. If $c$ is not null, $c$ is divided $\mathfrak{l}^{iv}$ per block. The pseudorandom keystream generator executes the entire blocks of $c$. Then, outputs $n$ bits $K$.

The design of SCKDF is allows combination of different types of pseudorandom keystream generator for computational extractor and pseudorandom expander (*Chuah, Dawson & Simpson, 2013*). This pseudorandom expander can generate arbitrary length of cryptographic key without discarding any excess bits, thus enhancing efficiency (*Canniere & Preneel, 2008*). However, pseudorandom keystream generator is not able to accommodate inputs of arbitrary length, the modification made to process the key derivation function inputs for the computational extractor phase has led to the SCKDF is not CAM-secure.

Definition 6. (PKG) (*Katz & Lindell, 2014*). The pseudorandom keystream generator is considered to pass all statistical tests within polynomial time if the polynomial time algorithm can distinguish between the output sequence of the generator and a truly random sequence of equal length with probability significantly greater than $\frac{1}{2}$.

## Block cipher based key derivation function

NIST SP 800-56C specified a cipher-based message authentication code based key derivation function (BKDF) (*Barker, Chen & Davis, 2020*) which consists of computational extractor and pseudorandom expander. The BKDF has been proven CAM-secure (*Koh & Chuah, 2020*). The computation extractor of BKDF is defined as:

$$PRK_i \leftarrow \mathfrak{B}_s\left(PRK_{i-1}\bigoplus D_i\right) \tag{8}$$

- $\mathfrak{B}$ is advanced encryption standard with key length of either 128 bits, 192 bits or 256 bits.
- $D_i$ is $p$ divided into 128 bits per block.
- $s$ is considered as the key for advanced encryption standard.
- Initial $PRK_0$ is $0^{128}$. $l^{PRK}$ is 128 bits.
- $1 \leq i < t, t = \left\lceil \frac{\mathfrak{l}^{p}}{128} \right\rceil$.

The pseudorandom expander of BKDF is defined as:

$$K(i) \leftarrow \mathfrak{B}_{PRK}\left(K_{i-1}\bigoplus M_i\right) \tag{9}$$

- $\mathfrak{B}$ is advanced encryption standard with key length of 128 bits.
- $M_i$ is considered as $c$ with 128 bits per block.
- $PRK$ is considered as the secret key for advanced encryption standard.
- Initial $K(0)$ is $0^{128}$.
- $1 \le i < t, t = \lceil \frac{l^c}{128} \rceil$,

If $n$ is greater than 128, the iterations in generating $K$ are continued until the required length is exceed by $\lceil \frac{n}{128} \rceil$. The $K$ is comprised of the initial $n$ bits, while the remaining bits are deleted.

The benefit of BKDF is its ability to handle inputs of arbitrary length. A similar drawback of HKDF is that it produces blocks of a fixed-length and discards any excess bits, leading to inefficiency. Another limitation is that the pseudorandom expander is fixed for advanced encryption standard with a key length of 128 bits.

# THE PROPOSED HMSKDF SCHEME

We formalized our key-hash based computational extractor (Eq. (4) satisfies Definition 5) and stream based pseudorandom expander (Eq. (7) satisfies Definition 6) which is relatively straightforward function as shown in Fig. 2. The computational extractor based HMAC_SHA512 which takes arbitrary length inputs of $p$ and $s$. The Trivium based pseudorandom expander generates arbitrary length of $n$ bits cryptographic key.

## Security analysis

This section provides a security analysis of general attacks that can be used against the different types of computation extractors and a formal security proof for the proposed HMSKDF scheme.

- ***Brute force attack and collision attack***

Assuming that no weaknesses are present in the key derivation functions, the key derivation functions are vulnerable to brute force attacks and collision attacks against the internal state of the computation extractor. If the internal state of the computation extractor is compromised, it can be used to generate the entire cryptographic keys. It should be noted that if the cryptographic keys are compromised, they can no longer be used to protect the security of electronic data.

The brute force attack is a method where the attacker systematically generates all possible strings of internal states of the computation extractor. The adversary can try all possible combinations of bits in the string until the correct one is found, which is then used to generate the cryptographic key. If the length of the internal state is $l^{is}$. Then, the complexity to brute force the internal state is $2^{l^{is}}$.

The collision attack is a method where the adversary uses the concept of the birthday paradox algorithm to find two or more inputs into the key derivation function that generate the same cryptographic key. With an internal state length of $l^{is}$. using the birthday paradox

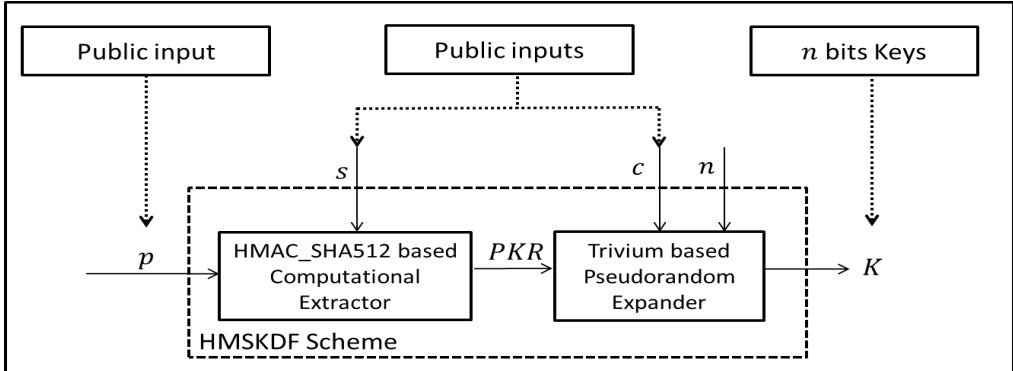

**Figure 2  The proposed HMSKDF Scheme.**

to calculate the collision, there is approximately a 50% chance of internal state collision at $2^{\frac{l^{is}}{2}}$.

Table 1 shows the finding of the brute force attack and collision attack towards the different type computation extractors. In general, extractors based on key-hash message authentication code (HMAC) for both HMAC_SHA1 and HMAC_SHA512, offer heightened security against brute force and collision attacks, compared to extractors based on stream ciphers or block ciphers. The extract with the lowest complexity is Trivium-based, with complexity of $2^{80}$ for brute force attack and $2^{40}$ for collision attack. Conversely, the extractor with the highest complexity is HMAC_SHA51-based, with complexity of $2^{512}$ for brute force attack and $2^{256}$ for collision attack. It can be inferred that the proposed key derivation function scheme (HMSKDF) with an HMAC-SHA512 based computation extractor and Trivium based pseudorandom expander offer high level of security with complexity of $2^{512}$ for brute force attack and $2^{256}$ for collision attack.

● *Formal security analysis*

HMSKDF is a two phases key derivation function scheme which constructed using the HMAC-SHA512 based computation extractor and Trivium based pseudorandom expander. As presented in 'The proposed HMSKDF Scheme', the HMAC-SHA512 based computational extractor and Trivium based pseudorandom expander (satisfying Definition 5 and Definition 6, respectively).

The following theorem establishes the security of HMSKDF based on the highest CAM security model. In this model, the polynomial adversary plays the CAM game and make at most $q$ queries. The polynomial adversary can win this indistinguishability game with probability not greater than $\frac{1}{2} + \varepsilon$, where $\varepsilon$ is deemed negligible. We now provide the formal security proof of the HMSKDF scheme.

Theorem 1: Suppose that a HMAC_SHA512 functions as an ideal pseudorandom function that satisfying Definition 5 and Trivium operates as an ideal pseudorandom keystream generator that satisfying Definition 6. If a HMSKDF is constructed using HMAC_512 based computational extractor and Trivium based pseudorandom expander,

**Table 1 Complexity of brute force attack and collision attack towards the different type computation extractors.**

| Computational extractor | Brute force | Collision |
|---|---|---|
| AES based extractor | $2^{128}$ | $2^{64}$ |
| Rabbit based extractor | $2^{128}$ | $2^{64}$ |
| Trivium based extractor | $2^{80}$ | $2^{40}$ |
| HMAC_SHA1 based extractor | $2^{160}$ | $2^{80}$ |
| HMAC_SHA512 based extractor | $2^{512}$ | $2^{256}$ |

the HMSKDF scheme is $(m, \min\{q_X, q_Y\}, \min\{t_X, t_Y\}, \varepsilon_X + \varepsilon_Y)$-CAM secure with the respect to the private string $p$ with entropy $m$.

Proof: To meet the requirements stated in Theorem 1, we must demonstrate: (a) the HMAC_512 based computational extractor is a $(m, q_X, t_X, \varepsilon_X)$-computational extractor; (b) the Trivium based pseudorandom expander is a $(q_Y, t_Y, \varepsilon_Y)$-pseudorandom expander.

To prove (a), we assume that the HMAC_SHA512 based computational extractor is not a $(m, q_X, t_X, \varepsilon_X)$-computational extractor. This would mean that the probability for all probabilistic polynomial time adversary $t_X$ can distinguish a $l^{PRK}$ bits string. The string can be $PRK$, where it is generated by HMAC_SHA512 based computational extractor using the inputs $p$ with $m$ entropy or a truly random string. The probability probabilistic polynomial time adversary makes the correct guess is not greater $\frac{1}{2} + \varepsilon_X$, where $\varepsilon_X$ is deemed negligible. This would also imply that the adversary can differentiate between the $PRK$ and a truly random string of the same length in the context of the underlying HMAC-PRF using a polynomial time method. This contradicts the assumption that HMAC_SHA512 functions as an HMAC-PRF satisfies Definition 5. Hence, the statement (a) is proven to be true.

To prove (b), we assume that Trivium based pseudorandom expander based is not a $(q_Y, t_Y, \varepsilon_Y)$-pseudorandom expander. This would mean that the probability for all probabilistic polynomial time adversary $t_Y$ can distinguish a $n$ bits string. The string can be $K$, where it is generated by Trivium based pseudorandom expander or a random string. The probability probabilistic polynomial time adversary makes the correct guess is not greater $\frac{1}{2} + \varepsilon_Y$, where $\varepsilon_Y$ is deemed negligible. This would also imply that the adversary can differentiate between $K$ and a truly random string of the same length in the context of the underlying Trivium pseudorandom keystream generator using a polynomial time method. This contradicts the assumption that Trivium as secure pseudorandom keystream generator satisfies Definition 6. Hence, the statement (b) is proven to be true.

Hence, by Theorem 1 the HMSKDF that is built from the HMAC_SHA512 based computation extractor and Trivium based pseudorandom expander is $(m, \min\{q_X, q_Y\}, \min\{t_X, t_Y\}, \varepsilon_X + \varepsilon_Y)$-CAM secure with the respect to the private string $p$ with $m$ entropy.

## Software performance evaluation

In this section, the software performance is showcased by analyzing the execution time of 25 different combinations of computational extractors and pseudorandom expanders. These combinations include HMAC_SHA1 (*Eastlake & Hansen, 2017*), HMAC_512 (*Eastlake*

**Table 2  Experiment parameters.**

|  | p | s | c | n |
|---|---|---|---|---|
| Experiment 1 | 128 bytes | 8 bytes | 32 bytes | 64 bytes |
| Experiment 2 | 256 bytes | 8 bytes | 32 bytes | 192 bytes |

& Hansen, 2017), AES128 (*Song et al., 2006*), Rabbit (*Boesgaard, Vesterager & Zenner, 2008*) and Trivium (*Robshaw, 2008*). In this simulation, one includes the existing key derivation function schemes such as hash-based message authentication code based key derivation (*Krawczyk, 2010*), block cipher based key derivation function (*Barker, Chen & Davis, 2020*) and stream cipher based key derivation function (*Chuah, Dawson & Simpson, 2013*).

There are two experiments were conducted to calculate the running time required to generate $n$ bits $K$ using the parameters $p$, $s$ and $c$. The lengths of the parameters $p$, $s$, $c$, and $n$ are based on the Host Identity Protocol (*Moskowitz et al., 2015*) as shown in Table 2. The experiments are simulated 100 times and the resulting times are recorded. The average simulation time is then calculated. The execution time is captured in nanoseconds using a CLOCK system. All the simulations were performed on a machine with the following specifications: AMD Ryzen 7 5700U with Radeon Graphics, 1.80 GHz, 16.0GB RAM and running a 64 bits Windows operating system.

Figure 3 displays the simulation results of the experiments, with experiment 2 featuring longer input lengths compared to experiment 1. For all key derivation functions schemes, the execution time shows an increase from experiment 1 to experiment 2.

The Trivium based key derivation function performs faster when the input length is shorter, taking just 13,815 nanoseconds. Conversely, it slows down as the input length $l^p$ increases (experiment 2). In comparison, the key derivation function using HMAC_SHA1 as the computational extractor and Trivium as the pseudorandom expander shows faster simulation speeds of 19,247 nanoseconds in experiment 2. On the other hand, the AES based key derivation function demonstrates the slowest simulation speeds among the different schemes. The combination of HMAC_SHA512 as the computational extractor and Trivium as the pseudorandom expander shows an average faster simulation time of 36,014 nanoseconds for experiment 1 and 41,618 nanoseconds for experiment 2.

## Hardware performance

Table 3 illustrates the hardware performance for hash functions (SHA1 and SHA512), stream ciphers (Trivium and Rabbit) and block cipher (AES). The results indicate that the AES has the lowest throughput, while Trivium requires fewer resources and has the highest throughput, and SHA512 has the second highest throughput at 2909 Mb/s. Overall, these results suggest that a hardware-based HMSKDF utilizing HMAC-SHA512 and Trivium offers notable medium throughput and efficiency compared to other key derivation function schemes.

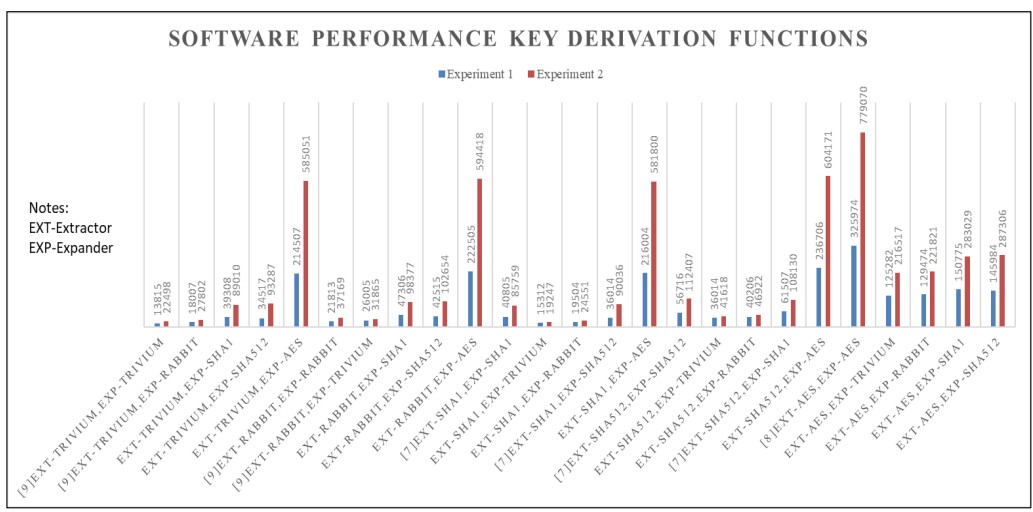

**Figure 3** Software performance (Time).

**Table 3 Complexity of ideal computational extractor based on stream ciphers, HMAC and block ciphers.**

|  | SHA1 (*Satoh & Inoue, 2007*) | SHA512 (*Satoh & Inoue, 2007*) | Trivium (*Good & Benaissa, 2007*) | Rabbit (*Boesgaard, Vesterager & Zenner, 2008*) | AES (*Satoh et al., 2001*) |
|---|---|---|---|---|---|
| Gates | 9859 | 27297 | 4921 | 28000 | 5398 |
| Technology (μm) | 0.13 | 0.13 | 0.13.13 | 0.18 | 0.11 |
| Throughputs (Mb/s) | 2006 | 2909 | 22300 | 473.6 | 311.09 |

# CONCLUSION

The key derivation function is an essential component in cryptographic systems, utilized to create cryptographic keys from non-uniformly random strings. This research examines different combinations of computational extractors and pseudorandom expanders in a cryptographic context, with emphasis on factors such as execution time, hardware performance and security analysis. It is crucial that the resulting cryptographic keys are impossible to differentiate from random binary strings of equivalent length, as they play a vital role in protecting data during storage and transmission across unsecured channels. Hence, this study includes different cryptographic primitives such as HMAC_SHA512, HMAC_SHA1, AES128, Trivium, and Rabbit. The findings indicate that the combination of a HMAC_SHA256 based computational extractor and a Trivium based pseudorandom expander is ideal for creating a secure key derivation function scheme, denoted as HMSKDF. The HMSKDF demonstrates the highest security with a brute force complexity of are $2^{512}$ anda collision attack complexity of $2^{256}$ as well as average efficiency in terms of execution time and throughput. Overall, this article establishes that the HMSKDF is proven $(m, q, t, \varepsilon)$-CAM secure and efficient as alternative key derivation function proposal that can be implemented in existing applications.

### Funding

This work was fully supported by Guangdong University of Science & Technology (Grant no. GKY2023BSQD-46). The funders had no role in study design, data collection and analysis, decision to publish, or preparation of the manuscript.

### Grant Disclosures

The following grant information was disclosed by the authors:
Guangdong University of Science & Technology: GKY2023BSQD-46.

### Competing Interests

The authors declare there are no competing interests.

### Author Contributions

- Chai Wen Chuah conceived and designed the experiments, performed the experiments, analyzed the data, prepared figures and/or tables, authored or reviewed drafts of the article, and approved the final draft.
- Nur Ziadah Harun performed the experiments, performed the computation work, prepared figures and/or tables, and approved the final draft.
- Isredza Rahmi A. Hamid performed the experiments, analyzed the data, authored or reviewed drafts of the article, and approved the final draft.

### Data Availability

 The source code files are available in the Supplemental Files.

### Supplemental Information

Supplemental information for this article can be found online at http://dx.doi.org/10.7717/peerj-cs.2249#supplemental-information.

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
