# Peer review of "Key derivation function: key-hash based computational extractor and stream based pseudorandom expander"

_PeerJ Computer Science, doi:10.7717/peerj-cs.2249_

## Round 0.1 · original submission · Major Revisions

Dear authors, the reviewers have proposed several additions and revisions to your article. Please address all comments and resubmit the article along with a response letter.

Reviewer 1 ·

Basic reporting

The article discusses a new approach to designing a key derivation function, which is a cryptographic algorithm used to generate cryptographic keys from private and public strings. The article includes theoretical definitions, security analysis, and performance evaluations of the proposed key derivation function scheme.
The organization of the article is generally good, and for the most part, the author keep the material in relevant sections. The level of English in the article is good, maintaining a professional tone and effectively conveying complex concepts in cryptography. All cited sources have a direct relevance to the topic of the article.

Experimental design

In general, the experimental design of this article requires some improvements for publication. Below I leave my suggestions for improving the article:
- In Section 1 (Introduction), the research gap and research question need to clarify and then justify it using Related Work Section for emerging applications. The authors contribution also needs to clarify, I think it need to write in bullet form.
- The manuscript lacks details about the software performance evaluation mentioned in Section 6.
- No methodological details are provided for the security analysis.

Validity of the findings

The article demonstrates scientific novelty and practical value. All underlying data have been provided, exhibiting robustness, statistical soundness, and proper control. The conclusions are clearly formulated.
Below I leave my suggestions for improving the article:
- The paper doesn't explicitly assess the impact of the proposed HMSKDF scheme. It would be beneficial to compare its security and efficiency with existing KDFs used in protocols.
- The paper mentions combining HMAC and a stream cipher for KDF is not novel, but it doesn't discuss existing literature on such approaches and how HMSKDF differs.

Additional comments

I recommend reviewing the article again after implementing improvements according to my recommendations.

Reviewer 2 ·

Basic reporting

The manuscript is generally well-structured with clear sections delineating the introduction, methods, results, and conclusions. The abstract provides a concise overview of the research and its significance. However, the manuscript suffers from an overabundance of well-known information, particularly in the first ten pages which largely reiterate established knowledge in the field. This could be significantly condensed to make way for more original contributions. Moreover, the presentation and readability of diagrams (e.g., Figure 3) need improvement as they are currently difficult to interpret, which detracts from the overall quality of the manuscript.

Experimental design

The experimental design outlined in the paper involves replacing basic cryptographic primitives in a classical key derivation function (KDF) with alternatives such as HMAC-SHA-512 combined with Trivium. While the idea is conceptually interesting, the justification for this specific combination is inadequately addressed. The paper lacks a clear rationale for selecting HMAC-SHA-512 and Trivium, especially given the tables provided do not indicate this combination as being the most efficient or secure. The choice seems arbitrary, which weakens the scientific rigor of the manuscript.

Validity of the findings

The findings of the manuscript claim to offer a high level of security and efficiency; however, these claims are not sufficiently substantiated by the data presented. The tables and comparisons in the paper fail to demonstrate a clear advantage of the proposed combination over existing methods, which questions the validity of the conclusions. Additionally, the metrics used for comparison (execution time of the algorithm) are influenced by many external factors. A more reliable metric, such as cycles per byte, would likely provide a more accurate and reproducible measure of efficiency and should be considered for future studies.

Additional comments

The paper would benefit from a clearer explanation of the choice of cryptographic primitives, especially considering the target applications. For high-security needs, alternatives like the STRUMOK stream cipher might offer a more appropriate comparison point than Trivium. Similarly, for low-resource environments, pairing Trivium with a hardware-oriented hash function like Quark might be more beneficial than using SHA-512.

·

Basic reporting

The article is presented in a logical way, overall well written and contains clear, unambiguous, technically correct text. The article conforms to professional standards of courtesy and expression. The text is clear and easy to read.
The article should include sufficient introduction and background to demonstrate how the work fits into the broader field of knowledge. The basis for the study objective was the 17 publications analysed in the initial two sections of the article and presenting by the authors the most relevant developments in the field. The references are relevant to the topic and cover both historical (prior) literature and most recent advancements.
The structure of the article in general conforms to the acceptable format of “standard sections”. Figures are relevant to the content of the article, of sufficient resolution and appropriately described and labelled. All appropriate raw data has been made available in accordance with the adopted Data Sharing policy.
The submission is ‘self-contained’, represents an appropriate ‘unit of publication’ and includes all results relevant to the hypothesis. Coherent bodies of work is not inappropriately subdivided merely to increase the number of publications.
All notations and terminology are clearly defined

Experimental design

The content of your article fits perfectly within in the scope of the journal PeerJ Computer Science. One reason for this is that a key topic of original primary research involves issues related to the development of key derivation function as an essential component in cryptographic systems. This is very important from the point of view of the essentiality of protecting electronic data transmitted over insecure channels.
The authors focused their study on solutions for improving the security and performance of the key derivation function. The submission clearly identified the research question and the knowledge gap under investigation and defined how the study contributes to filling it. To achieve this, an alternative approach to developing key derivation functions involving the combining of two different cryptographic ciphers under the title HMSKDF (key derivation function scheme) was proposed. This is relevant, meaningful and interesting because the article describes a new, previously unpublished contribution to the creation of cryptographic keys from non-uniformly random strings.
The investigation was conducted rigorously and in accordance with the prevailing technical and ethical standards in the field.
The presented methodology and its principles are reasonable and appropriate. The methods are described with sufficient information to replicate and can be reproduced by another investigator.

Validity of the findings

The research focused on security evaluation, includes validation and verification of software and hardware, namely comparisons of the performance of the software and hardware of the proposed alternative approach, as well as the complexity of brute force attack and collision attack towards the different type computation extractors. The authors proved that the results they obtained from the examination of different combinations of computational extractors and pseudorandom expanders in a cryptographic context can be considered satisfactory. Overall, it is established that the proposed HMSKDF proves itself as a secure and efficient scheme and is suitable for implementation in existing applications.
The underlying data on which the conclusion is based has been provided, and they are robust, statistically sound and controlled. The article contains some new data.
The conclusion is appropriately stated and its content is consistent with the evidences and arguments presented and addresses the main original question investigated.

Additional comments

Dear Authors,

Please address following minor concerns:

1. It would be best to clearly identify which of the previous works by all authors constitute the foundation of the work presented in this article.
2. The study considered the key derivation functions to brute force attack and collision attack. What were the other alternatives to the attacks?
3. There are parts of the article that can be:
3.1) compressed, namely section 2 can be shortened, for example by removing the explanation of the identical symbols occurring in the formulas (e.g. “...exclusive or” in formulas (4-7) , “...an ideal pseudorandom keystream generator [14]” in formulas (4) and (5)) etc.,
3.2) expanded, for instance section 3.
4. Please refer to the fact that the F symbol defined in the main text to denote the different components to make it challenging to read the content of the work and grasp the main idea. Take, for instance, F as “an ideal pseudorandom keystream generator [14]” (lines 178 and 190) and “advanced encryption standard” (lines 213 and 221).
5. To my mind, the use of the definition “most optimal” (8. Conclusion, line 341) is incorrect. It can only be “optimal”, while the use of the terminology “the most (more) optimal” or “the least (less) optimal” is questionable.
6. There are some editorial and linguistic errors, including punctuation mistakes in the article, please check and correct them. For example, For example, (i) separating 9 and 10 in [7, 9 – 10] (line 81), (ii) moving (9) from line 220 to line 219, and so on.

Best regards,
Reviewer.

---

## Round 0.2 · accepted · Accept

The reviewers have found the revision adequate for accepting the article for publication

Reviewer 1 ·

Basic reporting

The authors answered all my comments in the review, and modified the paper accordingly.

Experimental design

The authors answered all my comments in the review, and modified the paper accordingly.

Validity of the findings

The authors answered all my comments in the review, and modified the paper accordingly.

Additional comments

The authors answered all my comments in the review, and modified the paper accordingly.

·

Basic reporting

Dear Authors,

Thank you very much for taking my comments and suggestions into account. I am satisfied with your revised version of the article.

Best regards,
Reviewer.

Experimental design

Dear Authors,

Thank you very much for taking my comments and suggestions into account. I am satisfied with your revised version of the article.

Best regards,
Reviewer.

Validity of the findings

Dear Authors,

Thank you very much for taking my comments and suggestions into account. I am satisfied with your revised version of the article.

Best regards,
Reviewer.

Additional comments

Dear Authors,

Thank you very much for taking my comments and suggestions into account. I am satisfied with your revised version of the article.

Best regards,
Reviewer.